# Plant Diseases Identification through a Discount Momentum Optimizer in Deep Learning

**Yunyun Sun [1,\*], Yutong Liu [2], Haocheng Zhou [1] and Huijuan Hu [2,3]**

[1] School of Internet of Things, Nanjing University of Posts and Telecommunications, Nanjing 210023, China; 1018071607@njupt.edu.cn

[2] School of Computer Science and Technology, Nanjing University of Posts and Telecommunications, Nanjing 210023, China; 1219043635@njupt.edu.cn (Y.L.); hhj@njupt.edu.cn (H.H.)

[3] Institute of Network Security and Trusted Computing, Nanjing 210023, China

[\*] Correspondence: 2019070268@njupt.edu.cn

**Abstract:** Deep learning proves its promising results in various domains. The automatic identification of plant diseases with deep convolutional neural networks attracts a lot of attention at present. This article extends stochastic gradient descent momentum optimizer and presents a discount momentum (DM) deep learning optimizer for plant diseases identification. To examine the recognition and generalization capability of the DM optimizer, we discuss the hyper-parameter tuning and convolutional neural networks models across the plantvillage dataset. We further conduct comparison experiments on popular non-adaptive learning rate methods. The proposed approach achieves an average validation accuracy of no less than 97% for plant diseases prediction on several state-of-the-art deep learning models and holds a low sensitivity to hyper-parameter settings. Experimental results demonstrate that the DM method can bring a higher identification performance, while still maintaining a competitive performance over other non-adaptive learning rate methods in terms of both training speed and generalization.

**Keywords:** convolutional neural networks; non-adaptive; optimization; hyper-parameter; crop identification

## 1. Introduction

The outbreak of plant diseases is a threat to food production and security at the global scale. It can cause disastrous consequences for smallholder farmers representing 85% of the world's farms whose livelihoods depend on healthy crops [1]. In order to manage the detection and spread of plant diseases, several diagnostic protocols are developed in literature. However, challenges exist that prevent this kind of technology from being adopted in practice [2].

In previous research, a variety of generic machine learning (ML) methods are popularity in plant diseases identification including K-nearest neighbor (KNN), support vector machines (SVM), artificial neural networks (ANN), amongst others [3]. These methods are relatively successful under limited and constrained setups. However, these traditional machine learning methods have the problems of incomplete feature selection and fussy manual feature selection [4]. Deep Learning (DL) in particular offers very novel approaches to classify images because it extends classical ML by adding more "depth" (complexity) into the model [5]. These complex models can increase classification accuracy or reduce generalization error. The agricultural field, and especially the image-based plant diseases identification task, has not been an exception to this [6]. Indeed, since 2015, research on plant diseases detection has strongly veered towards using deep learning. AlexNet, GoogLeNet, VGG, ResNet, and DenseNet deep learning models are commonly used [7]. Solemane [8] identifies a mildew disease in crop millet and takes VGG16 as a pre-trained model with ImageNet as a source dataset. The method shows the effectiveness of transfer

learning for disease classification with small data. The performance of the method gives 95.00% accuracy. It is not suitable for the identification of other plant diseases. Within the plantvillage data set, Mohanty trains plant diseases identification model and achieves an accuracy of 99.35% [9]. However, when tested on images taken under conditions different from the images used for training, the model's accuracy is reduced substantially. In [10], the authors fine-tune pretrained deep convolutional neural networks of AlexNet, GoogLeNet, and VGGNet using the LifeCLEF 2015 plant task dataset.They have improved the overall validation accuracy of the top system by 15% points while outperforming the top three competition participants in all categories. In [11], fine-tuning and evaluation of state-of-the-art deep convolutional neural network for image-based plant disease classification are performed. DenseNets obtains a test accuracy score of 99.75% for the 30th epoch, beating the rest of the architectures. This research needs to be done to improve on the computational time and training process. As reported in [12], the authors focused on techniques to achieve an accuracy score of over 93% with class weight, SMOTE (Synthetic Minority Over-sampling Technique), and focal loss with deep convolutional neural networks from scratch. The goal was to counter high-class imbalance so that the model can accurately predict underrepresented classes. Their dataset biased towards Cassava Mosaic Disease and Cassava Brown Streak Virus Disease classes. They need further research though for multiple diseases on the same plant and multiple diseases on different plants.

Training CNNs (Convolutional Neural Networks) to achieve high multiple plant diseases identification accuracy was very challenging due to two reasons: (1) Deep learning is highly dependent on the dataset. However, there are few public data sets in the field of plant diseases identification. (2) The deep learning model has more network layers and parameters resulting in more time and cost for training and validation. (3) Existing methods only focus on single-target and few-target plant diseases image with simple backgrounds. In real life, crop diseases have the characteristics of various types, large quantity, and complex backgrounds. To solve the above problems, this paper studies the optimization method in deep learning towards plant identification tasks. The non-adaptive learning rate optimization method has been widely applied in deep learning, with the virtues of global optimization and rapid convergence. Based on the non-adaptive learning rate method, a new optimization algorithm is presented to increase the accuracy of identification. As a whole, the contribution of this article is as follows:

- Applying the discount weighted moving average to the momentum buffer $m_t$, a relative result reveals the higher recognition ability and faster convergence.
- Another key contribution of this work is show that DM does provide performance gains over other non-adaptive learning rate methods on plant diseases classification task.
- It is proved that discount momentum optimizer is insensitive to deep learning architectures and hyper-parameters.
- The DM method is capable of recovering popular non-adaptive learning rate methods in an efficient and accessible manner.

The rest of this paper is organized as follows: Section 2 introduces a state-of-the-art of deep learning optimization technology. Section 3 describes the details of non-adaptive learning rate methods as well as the proposed DM optimizer. Section 4 presents the implementation, empirical results, and analysis. The major work is discussed and wrapped up in Sections 5 and 6.

## 2. Related Work

Deep learning optimization methods are currently used to deal with the overfitting and performance deterioration problems. There are several common optimization methods. Here, we introduce the transfer learning method at first. As reported in [9–11,13], transfer learning techniques fine-tune transmitted sub-networks to adapt to new data and then mining depth features, which can effectively solve the small data sets problem. Another optimization method called data augmentation. This method enlarges the dataset to reduce the chance of over-fitting. Data enhancement methods include segmented symptom images,

geometrical transformations, and intensity transformations [10,14]. With the exception of the above-mentioned two optimization methods, optimizing network parameters of the deep learning models is also commonly used. These optimization methods improve the overall performance of models from convergence, over-fitting, running time, and generalization. A stochastic gradient descent (SGD) optimizer is one of the heavily used optimization methods in deep learning. Stochastic gradient descent (SGD) serves as a popular optimizer in deep learning. It is a non-adaptive learning rate method. That is, the learning rate needs to be manually determined. In [8,11,13–18], they have improved the validation accuracy performance on plant diseases identification tasks by employing the SGD optimizer. However, beyond that, k-fold cross-validation, batch normalization, and dropout also have a positive impact on the performance in deep learning model training. The k-fold cross-validation methods solve the over-fitting problems [12], and the batch normalization method potentially helps in two ways: faster learning and higher overall accuracy [11,12], and the dropout operation [18] prevents over-fitting and improves the generalization ability. Performance gains achieved by different methods highlight the important role of the optimization algorithm in deep learning.

## 3. Non-Adaptive Learning Rate Methods

This paper contributes to the plant diseases identification by investigating non-adaptive learning rate techniques. A typical deep learning optimization task consists of minimizing the objective function $f(\omega)$ and fixing the best set of parameters. Non-adaptive learning rate methods heavily utilized in optimization problems to update the weights are the workhorse in literature. Inspired by classical and successful gradient descent methods, we focus on the non-adaptive learning rate methods. Therefore, this paper provides an expansion and improvement of SGDM for a more general and robust CNN model. A generic framework of non-adaptive learning rate methods is shown in Algorithm 1. This enables us to understand the rules of non-adaptive learning rate methods.

---

**Algorithm 1** Generic framework of non-adaptive optimization methods

---

**Require:** $x_t \in \mathbb{R}^d$, initial step size (learning rate) $\alpha$, sequence of functions $\phi_t, \psi_t$
    **for** $t = 1$ to $T$
        $g_t = \nabla f_t(\omega_t)$
        $m_t = \phi(g_1, \ldots, g_t)$ and $V_t = I^2$
        $\eta_t = \alpha \cdot m_t$
        $\omega_t = \omega_{t-1} - \eta_t$
    **endfor**

---

Here, $\nabla f_t(\omega_t)$ is the gradient at $\omega_t$. For the sake of clarity, this paper summarizes the non-adaptive learning rate methods including Stochastic Gradient Descent (SGD) [19], Stochastic Gradient Descent with momentum (SGDM) [20], and Stochastic Gradient Descent with Nesterov momentum (NAG) [21] in Table 1. As observed in literature, there is a subtle difference between these non-adaptive learning rate methods in theoretical and implementation.

**Table 1.** An overview of non-adaptive learning rate methods.

| SGD | SGDM | NAG |
|:---:|:---:|:---:|
| $m_{t+1} = g_{t+1}$ | $m_{t+1} = \beta m_t + (1 - \beta)g_t$ | $g_{t+1} = \nabla f(\omega_{t+1} - \alpha \cdot m_t / \sqrt{V_t})$ |

### 3.1. Stochastic Gradient Descent Momentum

The momentum is a typical non-adaptive learning rate technique, like SGD, which can achieve optimal convergence guarantees. The momentum technique modifies the

SGD to accelerate convergence rate and to reduce oscillation. An update rule of SGD with momentum can be efficiently written as:

$$m_{t+1} = \beta \cdot m_t + (1 - \beta) \cdot g_t \tag{1}$$

$$\omega_{t+1} = \omega_t - \alpha m_{t+1} \tag{2}$$

where a new hyper-parameter $\beta \in [0,1)$ called the momentum parameter is an exponential discount factor. It determines how quickly the momentum buffer $m_t$ is updated and the variance of a normalized momentum buffer.

In SGDM, the update rule can also be written as:

$$\omega_{t+1} = \omega_t - \alpha\left[(1 - \beta) \cdot \sum_{i=0}^{t} \beta^i \cdot g_{t-i}\right] \tag{3}$$

**Definition 1.** *For $\beta \in (0,1)$, this paper defines the exponential discount function $\delta_{EXP,\beta}$ as:*

$$\delta_{EXP,\beta}(i) = (1 - \beta)\beta^i \tag{4}$$

**Definition 2.** *For a discount function $\delta$ and a sequence of vectors $x \in \mathbb{R}^d$, we define a discounted sum $DS_\delta(x)$ as:*

$$DS_\delta(x) = \sum_{i=0}^{t} \delta(i) \cdot x_{t-i} \tag{5}$$

*when $\sum_{i=0}^{t} \delta(i) = 1$ for all $t \geq 0$; this paper calls it a discounted sum average, and the exponentially weighted moving average $EWMA_\beta(x)$ is:*

$$EWMA_\beta(x) = DS_{\delta_{EXP,\beta}}(x) = (1 - \beta) \cdot \sum_{i=0}^{t} \beta^i \cdot x_{t-i} \tag{6}$$

EWMA can be viewed as a weighted average method to estimate the expectation of random variable $x = x_0 \dots x_t$. The theoretical above indicates that the momentum buffer $m_t$ is precisely an exponentially weighted moving average, viz., $m_t = EWMA_\beta(\nabla \ell_{0\dots t}(\omega_{0\dots t}))$.

*3.2. The Proposed Method: Discount Momentum Optimizer*

Inspired by EWMA, this paper extends the SGDM method to provide a distinct improvement in performance. The proposed algorithm can be regarded as a simple modification of the SGDM. Here, the details of the modifies are illustrated as follows:

**Definition 3.** *Similarity, the equation of the proposed discount function $\delta_{DM,v,\beta}(i)$ and the discount weighted moving average $DWMA_{DM,v,\beta}(x)$ are shown as follows:*

$$\delta_{DM,\mu,\lambda}(i) = (1 - \lambda)\lambda^i = \begin{cases} 1 - \mu\lambda & i = 0 \\ \mu(1 - \lambda)\lambda^i & i > 0 \end{cases} \tag{7}$$

$$DWMA_{DM,\mu,\lambda}(x) = DS_{\delta_{DM,\mu,\lambda}}(x)$$

$$= (1 - \mu) \cdot x_0 + \mu(1 - \lambda)\sum_{i=0}^{t} \lambda^i \cdot x_{t-i} \tag{8}$$

*where discount momentum hyper-parameters $\mu \in \mathbb{R}$ and $\lambda \in [0,1)$ are constant.*

Apply the update rule to the proposed algorithm:

$$\omega_{t+1} = \omega_t - \alpha[(1 - \mu\lambda) \cdot g_t + \mu(1 - \lambda) \sum_{i=1}^{t} \lambda^i \cdot g_{t-i}] \tag{9}$$

Like SGDM, the update rule can also be equivalently written as

$$m_{t+1} = \lambda \cdot m_t + (1 - \lambda) \cdot g_t \tag{10}$$

$$\omega_{t+1} = \omega_t - \alpha \cdot [(1 - \mu) \cdot g_t + \mu \cdot m_{t+1}] \tag{11}$$

This suggested that the proposed discount momentum (DM) method is a simple modification of exponentially weighted moving average. On the condition of momentum hyper-parameter $\mu = 1$, the discount momentum (DM) is precisely the SGDM.

## 4. Results

In this section, we present our empirical study on the performance of DM method and compare it with other non-adaptive learning rate methods on plant diseases identification tasks in terms of training performance and generalization. We separate experiments into those with hyper-parameter tuning and those with CNN architectures. In the experiment, training occurs over 90 epochs (minibatch size 64). We apply the learning rate decay schedule by a factor of 0.1 at 30 epochs' stepsize, which is commonly used in literature [22]. Each training run uses dual GPU (2*RTX 2080Ti).

### 4.1. The Dataset

A publicly-available and well-known database, plantvillage, is used for the training and testing of CNNs models. The plantvillage dataset contains 54,306 color leaf images with a uniform background and has 38 crop-disease pairs. These 38 classes comprise 14 crop plants and 26 different healthy or diseased plants. Some randomly selected images are shown in Figure 1. In our study, the images are divided into train and test subsets in an 80/20 ratio. It means that the training set contains 80% (43,810 images) of the total images and the remaining 20% (10,495 images) are used for the test data. In these non-adaptive learning rate approaches, we perform both model training and parameters' optimization on these images.

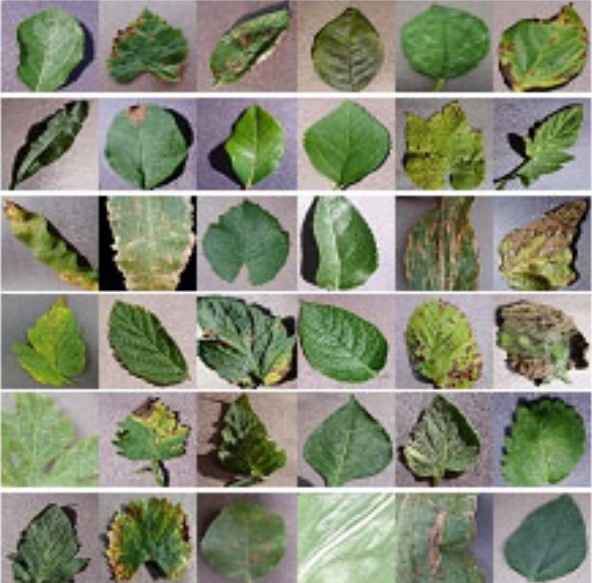

**Figure 1.** Randomly selected images from the plantvillage dataset.

### 4.2. Hyper-Parameter Tuning

Hyper-parameter tuning has a great influence on the quality of optimization for deep convolutional neural networks [23]. In this section, we discuss the discount momentum hyper-parameters $\mu$ and $\lambda$ sensitivity in an image classification task. We set $\mu \in \{0.0, 0.9\}$ and $\lambda \in \{0.9, 0.99, 0.999\}$ [24]. Generalization error under several hyper-parameters setting are presented in Figure 2. As observed in sensitivity experiments, there is little difference between these discount momentum hyper-parameters $\mu$ and $\lambda$ settings. Therefore, DM holds a low sensitivity to hyper-parameters.

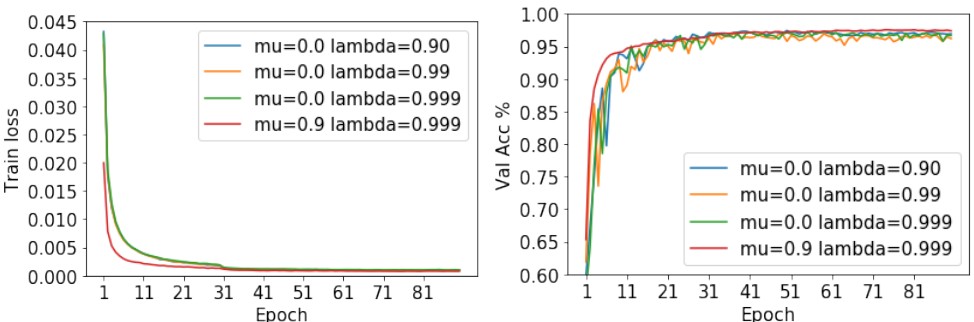

**Figure 2.** Hyper-parameters $\mu$ and $\lambda$ sensitivity experiments with DenseNet121 on the plantvillage dataset.

### 4.3. Convolutional Neural Networks

In the experiment, the DM method is applied to a variety of models. ResNet and DenseNet are typical convolutional neural networks' architectures, which are efficient and widely-used in literature. We consider testing the task of plant diseases classification with 50-layer ResNet and the 121-layer DenseNet. We select DM as the baseline algorithm and include comparisons with SGD, SGDM, and NAG non-adaptive learning rate methods. For DM, SGD, SGDM, and NAG, the first 30 epochs use learning rate $\alpha = 1.0$, the next 30 epochs use $\alpha = 0.1$, and the final 30 epochs use $\alpha = 0.01$. For SGDM and NAG, the momentum parameter $\beta$ is directly applied to default value 0.9.

**Plantvillage-ResNet50** ResNet50 has 50 layer deep CNNs with skip connections for image classification. We test our algorithm with the ResNet50 model on the plantvillage dataset. We compare the performance of DM, SGD, SGDM, and NAG. The results are shown in Figure 3, from which we can see that the DM algorithm is significantly better than SGDM and NAG. We notice that the training speed and generalization performance of DM are relatively superior SGD at the initial 30 epochs. In the later, DM and SGD share competitive results, while DM is generally slightly better.

**Plantvillage-DenseNet121** DenseNet121 is a 121-layer deep CNNs with dense connections. Results of this experiment are reported in Figure 3. As is expected, the overall performance of each algorithm on ResNet50 is similar to that on DenseNet121. We can see that the DM method performs better than the non-adaptive ones in training. In addition, compared with non-adaptive learning rate methods, it converges as fast as SGD and achieves a bit higher accuracy on the plant diseases identification task.

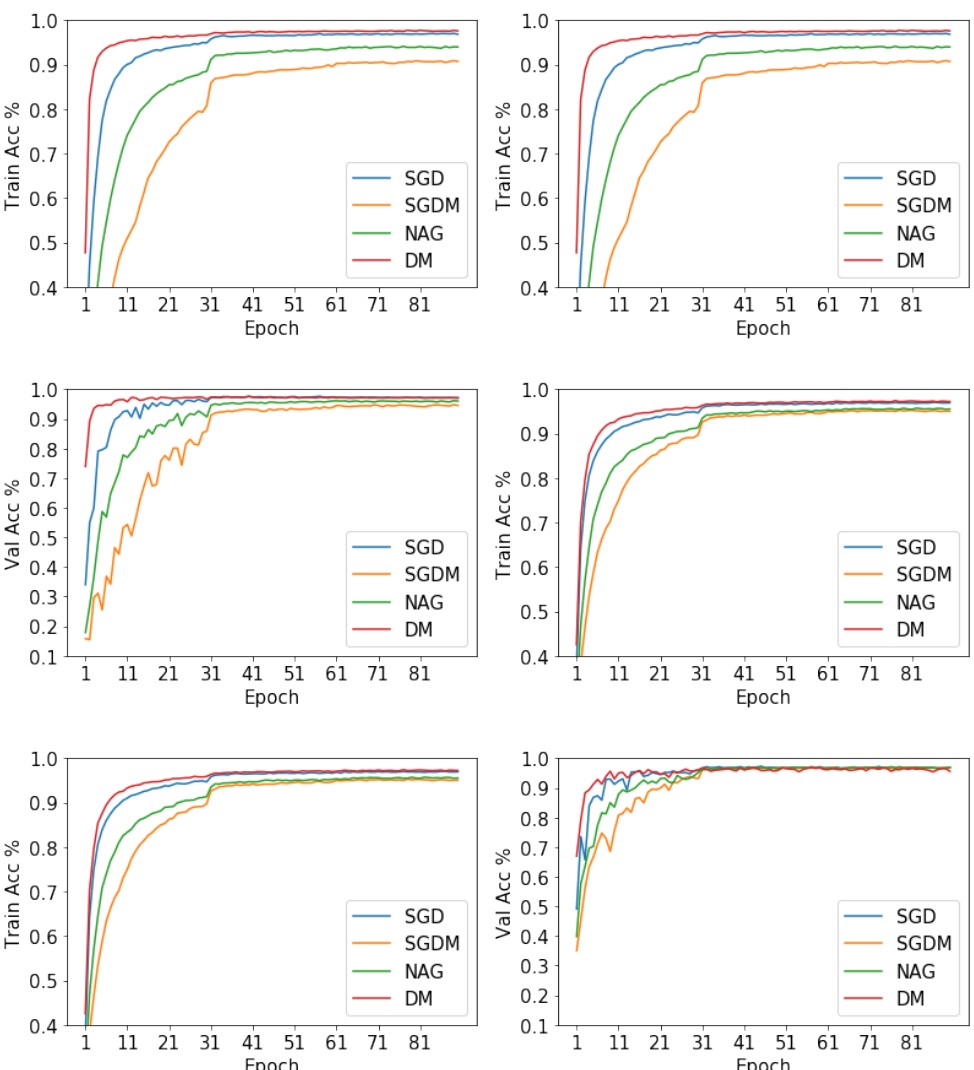

**Figure 3.** Additional empirical results on convolutional neural networks. (The first figure on the left is ResNet50-plantvillage train accuracy; the first figure on the right is ResNet50-plantvillage train loss; the second figure on the left is ResNet50-plantvillage validation accuracy; the second figure on the right is Densenet121-plantvillage train accuracy; the third figure on the left is Densenet121-plantvillage train loss; and the third figure on the right is Densenet121-plantvillage validation accuracy, respectively).

## 5. Discussion

With the widespread use of deep learning solutions in plant diseases identification tasks, some limitations on model training have been highlighted. These issues mainly include plant diseases images in different categories are unevenly distributed, the diversity of images is low, and the complexity of the network model is increased. These problems caused an increased training time, poor convergence, generalization performance, and low recognition accuracy, which restricts the popularization of deep learning solutions in disease recognition tasks. Thereby, the thesis undertakes a study on the network parameter optimization aspect and proposes a DM method to enhance the overall performance of CNNs. Actually, the proposed algorithm, referred to as discount momentum, is a variant of the SGDM method. Compared with the results obtained in DM with the current state-of-the-art non-adaptive learning rate algorithms, the DM method has improved the performance of the model in identification accuracy, convergence speed, and parameter sensitivity. In addition, DM can recover SGD, SGDM, and NAG methods by assigning parameters.

Here, we discuss the updated rules of these non-adaptive learning rate methods and their relationship with DM. The first is the SGD optimization algorithm. The SGD optimizer is heavily used in deep learning and performs well across image recognition domains in spite of its simplicity [25]. It considers mini batches to compute the unbiased estimate of the expected gradient. At each iteration, $m_t = g_t$. The network parameters are updated by

$$\omega_{t+1} = \omega_t - \alpha \cdot g_{t+1} \tag{12}$$

where, when the discount momentum hyper-parameter $\mu = 0$, the updates of parameter $\omega_t$ in the DM method are precisely (12). Therefore, the DM method can recover the SGD method with $\mu = 0$.

Next, we have the SGDM algorithm mentioned in Section 3.1. The details are shown in Section 3.1. Comparing the update rules of DM and SGDM, it is not difficult to find that, when the discount momentum hyper-parameter $\mu = 1$, the DM optimizer is precisely the SGDM method.

Last but not the least is the NAG algorithm. NAG is provided as a variant of the SGDM method. It can achieve a global convergence rate for general smooth convex functions. NAG takes inspiration from Nesterov's accelerated gradient method and slightly ahead in the measure of loss function gradient of the momentum [26]. In fact, NAG replaces the $m_t$ term of SGDM by using the $[(1 - \lambda) \cdot g_{t-1} + \lambda \cdot m_t]$. Therefore, the parameters are updated by

$$\omega_{t+1} = \omega_t - \alpha[(1 - \lambda) \cdot g_t + \lambda \cdot m_{t+1}] \tag{13}$$

The DM method recovers the NAG method with discount momentum hyper-parameter $\mu$ equal to hyper-parameter $\lambda$.

## 6. Conclusions

On the basis of an analysis of theoretical and experiments, we provide overwhelming evidence to the claim that the proposed algorithm is feasible to spread in deep learning fields. In non-adaptive learning rate methods, there are difficulties in the selection of hyper-parameters. This caused poor model performance and training to be difficult. We discussed the hyper-parameters setting and found that there is little accuracy performance change. Therefore, the DM algorithm is less sensitive to the change of hyper-parameters. We also discussed the adaptability of the DM method in different deep learning models. ResNet and DenseNet are typical convolutional neural networks architectures, which are representative in network layers and model architecture. We considered testing the task of plant diseases classification with 50-layer ResNet and 121-layer DenseNet. Results showed that DM has higher accuracy and is independent of the model, which is superior to state-of-the-art non-adaptive learning rate methods. We hope it is useful for the development of smart agriculture. Further studies should be needed to verify the applicability of the proposed algorithm in field experiments. In the future, we hope to integrate the proposed method into the mobile client and apply it to the field environment.

**Author Contributions:** Conceptualization, Y.S.; methodology, Y.S.; validation, Y.L. and H.Z.; investigation, Y.L.; data curation, Y.L. and H.H.; writing—original draft preparation, Y.S. All authors have read and agreed to the published version of the manuscript.

**Funding:** The subject is sponsored by the National Natural Science Foundation of P. R. China (No. 61872196, No. 61872194, No. 61902196, No. 62102194 and No. 62102196), Scientific and Technological Support Project of Jiangsu Province (No. BE2019740, No. BK20200753 and No. 20KJB520001), Major Natural Science Research Projects in Colleges and Universities of Jiangsu Province (No. 18KJA520008), Six Talent Peaks Project of Jiangsu Province (No. RJFW-111), Postgraduate Research and Practice Innovation Program of Jiangsu Province (No. KYCX19_0909, No. KYCX19_0911, No. KYCX20_0759, No. KYCX21_0787, No. KYCX21_0788 and No. KYCX21_0799).

**Institutional Review Board Statement:** Not applicable.

**Informed Consent Statement:** Not applicable.

**Data Availability Statement:** Not applicable.

**Conflicts of Interest:** The authors declare no conflict of interest.

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
