# Peer review of "Plant Diseases Identification through a Discount Momentum Optimizer in Deep Learning"

_applsci, doi:10.3390/app11209468_

Round 1
Reviewer 1 Report
Plant diseases identification through discount momentum optimizer in deep learning
General Comments
This study aims to explore an innovative optimization technique for plant diseases identification task to enhance the overall performance of CNNs. The results show that discount momentum (DM) deep learning optimizer results in obtaining higher recognition ability and faster convergence. Firstly, it is not clear if there is any substantial contribution from this work. This is a case study and the results are not interesting for the international readers. The literature review was not properly addressed, and the authors have failed to clearly explain the importance of this, as well as to point out which is the substantial contribution of this paper. In general, it is not clear either the relevance, motivation or novelty introduced by their study. An in-depth analysis of the results is missing, and the authors should further develop critical appraisal in their discussion. This paper in the present form is not suitable for publication due to lacks in new findings. In general, the study was poorly developed, and at this point, it does not meet the journal's standards.
Please see the individual comments to each section below.
Introduction
The Introduction section should be entirely rewritten and restructured. The literature review should be considerably improved, and the authors should clearly explain the relevance of this work in relation to the current state-of-the-art, as well as to point out which is the substantial contribution of this paper. The authors are also advised to reason both the main contributions and novelties introduced in their study.
Problem Statement
The problem of interest should be clearly defined in a new short section after the Introduction (problem statement or materials and methods).
Results and discussion
The discussion section should be considerably improved. The authors should further develop critical appraisal in their discussion. In addition, the authors are advised to use qualitative and/or quantitative reasoning to explain/discuss their results. In addition, it could be more interesting to compare the results obtained in this work with other studies available in the literature.
Conclusions
The Conclusions section is too short and it should be completely rewritten. The authors do not clearly summarize the actions taken and results.
In addition to summarizing the actions taken and main results, the authors should include an explanation about the significance of their results using quantitative reasoning in comparison with suitable benchmarks, mainly those from other studies presented in the literature.
The authors should include a clear and concise paragraph about their research limitations and future.
Author Response
Comments from the Editorial Office:
#Reviewer1
-General Comments
This study aims to explore an innovative optimization technique for plant diseases identification task to enhance the overall performance of CNNs. The results show that discount momentum (DM) deep learning optimizer results in obtaining higher recognition ability and faster convergence. Firstly, it is not clear if there is any substantial contribution from this work. This is a case study and the results are not interesting for the international readers. The literature review was not properly addressed, and the authors have failed to clearly explain the importance of this, as well as to point out which is the substantial contribution of this paper. In general, it is not clear either the relevance, motivation or novelty introduced by their study. An in-depth analysis of the results is missing, and the authors should further develop critical appraisal in their discussion. This paper in the present form is not suitable for publication due to lacks in new findings. In general, the study was poorly developed, and at this point, it does not meet the journal's standards.
Reply: We thank the Referee for the comments.
1-Reviewer’s Comments: Introduction
The Introduction section should be entirely rewritten and restructured. The literature review should be considerably improved, and the authors should clearly explain the relevance of this work in relation to the current state-of-the-art, as well as to point out which is the substantial contribution of this paper. The authors are also advised to reason both the main contributions and novelties introduced in their study.
Reply: We thank the Referee for suggesting these. We have rewritten and restructured the statement introduction as explained above. The revised version is on pages 1-3.
2- Reviewer’s Comments: Problem Statement
The problem of interest should be clearly defined in a new short section after the Introduction (problem statement or materials and methods).
Reply: We thank the Referee for pointing this out. We agree that “The problem of interest should be clearly defined in a new short section after the Introduction”. We have added the following sentences to the second paragraph of page 2.
“Training CNNs (Convolutional Neural Networks) to achieve high multiple plant diseases identification accuracy was very challenging due to two reasons: 1) Deep learning is highly dependent on dataset. However, there are few public data sets in the field of plant diseases identification. 2)Deep learning model has more network layers and parameters resulting in more time and cost for training and validation. 3) Existing methods only focus on single-target and few-target plant diseases image with simple backgrounds. In real life, crop diseases have the characteristics of various types, large quantity and complex backgrounds.”
3- Reviewer’s Comments: Results and discussion
The discussion section should be considerably improved. The authors should further develop critical appraisal in their discussion. In addition, the authors are advised to use qualitative and/or quantitative reasoning to explain/discuss their results. In addition, it could be more interesting to compare the results obtained in this work with other studies available in the literature.
Reply: We have properly modified the discussion section. The revised version is on pages 6-8.
4- Reviewer’s Comments: Conclusions
The Conclusions section is too short and it should be completely rewritten. The authors do not clearly summarize the actions taken and results.
In addition to summarizing the actions taken and main results, the authors should include an explanation about the significance of their results using quantitative reasoning in comparison with suitable benchmarks, mainly those from other studies presented in the literature.
The authors should include a clear and concise paragraph about their research limitations and future.
Reply: We have rewritten the conclusion section. The revised version is on page 8.
“On the basis of an analysis of theoretical and experiments, we provide overwhelming evidence to the claim that the proposed algorithm is feasible to spread in deep learning fields. In non-adaptive learning rate methods, there are difficulties in the selection of hyper-parameters. This caused poor model performance and training difficult. We discussed the hyper-parameters setting and found that the accuracy performance change is little. Therefore, DM algorithm is less sensitive to the change of hyper-parameters. We also discussed the adaptability of DM method in different deep learning models. ResNet and DenseNet are typical convolutional neural networks architectures, which are representative in network layers and model architecture. We considered test the task of plant diseases classification with 50-layer ResNet and 121-layer DenseNet. Resulted showed that DM has higher accuracy and is independent of model, which is superior to state-of-the-art non-adaptive learning rate methods.
We hope it is useful for the development of smart agriculture. Further studies should be needed to verify the applicability of the proposed algorithm in field experiments. In the future, we hope to integrate the proposed method into the mobile client and apply it to field environment.”

Reviewer 2 Report
No comments.
Author Response
We thank the Referee for the positive comments.
Reviewer 3 Report
The method mentioned shows an effective approach in the identification of plant disease. However, there are a few issues in this manuscript that have been mentioned in the following comments.
The English language needs to be improved. It is advised that the help of necessary tools should be taken in this regard.
line 25- reference needed to prove that "using ANN /SVM/ KNN... many of the difficulties associated with the intrinsic characteristics of the problem
26 can not be properly handled."
line 117- reference is needed for the reasoning behind chosen values..
line 126- mention the name/details for the random selection method of the leaf. it is crucial as it is the basis of your study and randomness choice affects (it should not though) the results.
text font size in all the plots should be increased and the line thickness should be enhanced for easier comprehension of the data/results for perspective readers.
No pictorial representation/scheme of this proposed method has been given in this manuscript to show the neural networks. Schemes are best and very popular to compare methods and easier understanding of ML/DL methods. addition of a neural network scheme is highly advised.
Author Response
Comments and Suggestions for Authors
1- Reviewer’s Comments:
The method mentioned shows an effective approach in the identification of plant disease. However, there are a few issues in this manuscript that have been mentioned in the following comments.
Reply: We thank the Referee for the comments.
2- Reviewer’s Comments:
The English language needs to be improved. It is advised that the help of necessary tools should be taken in this regard.
Reply: We have carefully checked the grammar in full text with help of professionals.
3- Reviewer’s Comments:
line 25- reference needed to prove that "using ANN /SVM/ KNN... many of the difficulties associated with the intrinsic characteristics of the problem cannot be properly handled."
Reply: We have modified the above sentences as the following sentences and added the reference ‘4’ in the second paragraph of page 1.
“However, these traditional machine learning methods have the problems of incomplete feature selection and fussy manual feature selection.”
4- Reviewer’s Comments:
line 117- reference is needed for the reasoning behind chosen values.
Reply: We have added the reference ‘26’ in the first paragraph of section experiments.
5- Reviewer’s Comments:
line 126- mention the name/details for the random selection method of the leaf. it is crucial as it is the basis of your study and randomness choice affects (it should not though) the results.
Reply: We thank the Referee for suggesting these. In experiment, we randomly selected 80% from each category as the training set and selected 20% as the test set.
6- Reviewer’s Comments:
text font size in all the plots should be increased and the line thickness should be enhanced for easier comprehension of the data/results for perspective readers.
Reply: We have enlarged the text font in all plots.
7- Reviewer’s Comments:
No pictorial representation/scheme of this proposed method has been given in this manuscript to show the neural networks. Schemes are best and very popular to compare methods and easier understanding of ML/DL methods. addition of a neural network scheme is highly advised.
Reply: We thank the Referee for pointing this out. In this paper, we study the optimization method in deep learning towards plant identification tasks. Based on non-adaptive learning rate method, a new optimization algorithm is presented to increase the accuracy of identification. We introduced state-of-the-art non-adaptive learning rate methods. We also discussed the adaptability of DM method in different deep learning models ResNet and DenseNet are typical convolutional neural networks architectures, which are representative in network layers and model architecture. We considered test the task of plant diseases classification with 50-layer ResNet and 121-layer DenseNet. Result showed that DM has higher accuracy and is independent of model.

Round 2
Reviewer 3 Report
comments seem addressed
Author Response

(The authors gave the same response as above.)
